# The Short and Intensive Rehabilitation (SHAiR) Program Improves Dropped Head Syndrome Caused by Amyotrophic Lateral Sclerosis: A Case Report

**DOI:** 10.3390/medicina58030452

**Published:** 2022-03-21

**Authors:** Ryunosuke Urata, Tatsuya Igawa, Akifumi Suzuki, Yutaka Sasao, Norihiro Isogai, Haruki Funao, Ken Ishii

**Affiliations:** 1Department of Orthopaedic Surgery, School of Medicine, International University of Health and Welfare, Chiba 286-8520, Japan; ryu.urata62@gmail.com (R.U.); sb11033.1028@gmail.com (A.S.); sasaospine@marianna-u.ac.jp (Y.S.); n.isogai0813@gmail.com (N.I.); 2Department of Orthopaedic Surgery, School of Medicine, International University of Health and Welfare Narita Hospital, Chiba 286-8520, Japan; 3Department of Orthopaedic Surgery, Spine and Spinal Cord Center, International University of Health and Welfare Mita Hospital, Tokyo 108-8329, Japan; 4Department of Rehabilitation, International University of Health and Welfare Mita Hospital, Tokyo 108-8329, Japan; 5Department of Physical Therapy, School of Health Science, International University of Health and Welfare, Tochigi 323-8501, Japan

**Keywords:** dropped head syndrome (DHS), amyotrophic lateral sclerosis (ALS), rehabilitation, short and intensive rehabilitation (SHAiR) program

## Abstract

*Background and Objectives:* Dropped head syndrome (DHS) is a syndrome that presents with correctable cervical kyphotic deformity as a result of weakening cervical paraspinal muscles. DHS with amyotrophic lateral sclerosis (ALS) is a relatively rare condition, and there is no established treatment. This is the first case report describing the improvement of both dropped head (DH) and cervical pain after the short and intensive rehabilitation (SHAiR) program in an ALS patient with DHS. *Case Report:* After being diagnosed with ALS in June 2020, a 75-year-old man visited our hospital in October 2020 to receive treatment for DHS. At the initial visit, the patient’s DH was prominent during standing and walking. The pain intensity of the neck was 9 out of 10 on the numerical rating scale (NRS), which was indicative of severe pain. The patient was hospitalized for 2 weeks and admitted into the SHAiR program. DH began to decrease one week after undergoing the SHAiR program and improved two weeks later. Neck pain decreased from 9 to 6 on the NRS. *Results:* The SHAiR program is a rehabilitation program aimed at improving DH in patients with idiopathic DHS. The program was designed to improve neck extensor and flexor function and global spinal alignment, and the program may have contributed to the improvement of DH and neck pain. Currently, reports of conservative therapies for this disease are limited to the use of cervical orthosis. Although further research is needed on the safety and indications of treatment, the SHAiR program may be a viable treatment option.

## 1. Introduction

Dropped head syndrome (DHS) presents as a cervical kyphotic deformity called chin-on-chest deformity that results from a weakening of the cervical paraspinal muscles [1]. DHS develops for a variety of reasons, including Parkinson’s disease (PD), myasthenia gravis (MG), and amyotrophic lateral sclerosis (ALS), in addition to idiopathic DHS (i.e., of an unknown cause) [2]. Previous studies have reported a very low prevalence of ALS with DHS [2,3,4]. Therefore, there are few reports of treatment for this disease. The short and intensive rehabilitation (SHAiR) program is a rehabilitation program developed to improve the dropped head (DH) of patients with idiopathic DHS [5]. This report presents a case of ALS in which the SHAiR program was effective. This patient showed improvement in DH and cervical pain in just 2 weeks. This case report was conducted and reported in accordance with the CAse REport (CARE) guidelines for reporting clinical cases.

## 2. Case Report

### 2.1. Patient Information

A 75-year-old male patient with ALS presented with DHS. In October 2019, the patient noticed difficulty swallowing and discomfort in the neck. After January 2020, the difficulty swallowing further progressed and he gradually became unable to support the head. The patient was diagnosed with probable bulbar-type ALS according to the Awaji criteria after undergoing a detailed examination at several hospitals in June 2020. The patient’s clinical findings at that time are shown in Table 1.

After the diagnosis of ALS, the DH, neck pain, speech difficulties, and dysphagia further progressed. For the treatment, medication (Riluzole, Edaravone, non-steroidal anti-inflammatory drugs [NSAIDs]) and conventional rehabilitation were prescribed. Although posture modification and daily walking (4000 steps per day) were provided by a physical therapist for approximately two months with conventional rehabilitation, the patient’s symptoms did not improve. The patient sought to alleviate the daily suffering caused by DH and visited our hospital for additional treatment. The patient’s medical history included ulcerative colitis, dyslipidemia, lumbar canal stenosis, and Bell’s palsy.

### 2.2. Clinical Findings and Diagnosis

The patient was referred to our hospital in October 2020 and was diagnosed with DHS caused by ALS by a specialist. The revised ALS functional rating scale (ALSFRS-R) score at that time was 38 out of 48 points. On initial physical examination, the symptoms were prominent during standing and walking. A therapist was able to easily lift the head of the sitting patient. This assessment determined that the patient’s pathology was not a consequence of cervical spine deformity with joint contracture. The patient experienced difficulty standing while maintaining a horizontal gaze (Figure 1a,b). In addition, the head during walking was always drooping. Thus, the patient was unable to follow the specialist’s instructions to “walk with your head raised” (Appendix A). The pain intensity of the neck was 9 out of 10 on the numerical rating scale (NRS), which was indicative of severe pain. Magnetic resonance imaging showed a thickening of the ligamentum flavum and posterior longitudinal ligament between the vertebral bodies of C5-6, C6-7, and C7-T1. However, there were no abnormal signal changes in the cervical spinal cord. Fasciculation was observed in the patient’s right lower extremity. Neck extension, right ankle dorsiflexion, and plantar flexion muscle strength measured by manual muscle testing (MMT) were grades 2, 3, and 3, respectively. Other muscular strength outcomes were normal (grade 5). The patient complained of numbness localized in the soles of both feet. In addition, the voice function and the speech intelligibility were also declining. The patient’s tongue had atrophied, thus written conversation was the primary means of communication. Swallowing function was also severely impaired and the patient had difficulty swallowing saliva. Although a gastrostomy was planned prior to visiting our hospital, the procedure was not performed due to poor bowel placement; therefore, the patient was eating finely chopped meals.

### 2.3. Therapeutic Intervention

To improve the patient’s symptoms, a prescription for the SHAiR program was suggested. The SHAiR program is a short-term intensive exercise program that includes thoracolumbar physical training to improve global spinal alignment, in addition to strength training of the neck extensors and flexors. Although an approach to improving neck function and correcting posture is often taken in conventional rehabilitation, a structured treatment program has not been established globally. The SHAiR program was considered the most suitable program for the patient’s treatment, because it has been reported with good short-term results and is a non-invasive treatment. The structure of the program is cervical paraspinal muscle exercise, deep cervical flexor muscle exercise, range of motion exercise with cervical and thoracic mobilization, hip lift exercise, anterior pelvic tilt exercise, and walking exercise. Although some of the training has been reported to be performed in the supine position [5], training in this position was considered a risk for his saliva aspiration. Therefore, all exercises performed in the supine position were replaced by Fowler’s position (Figure 2A). For the deep cervical flexor muscle exercise, a towel was used in place of the pressure biofeedback unit (Figure 2B,C). The number of sets and repetitions for each training session were individually adjusted by the therapist assessing patient fatigue and respiratory status.

The individualized SHAiR program was conducted by a physical therapist and an occupational therapist, one session each day, six times a week for two weeks. The patient received 20 min/day of speech therapy in addition to the SHAiR program. Furthermore, SHAiR program-based voluntary training and walking were also added by the patient. Voluntary walking was gradually implemented in the range of approximately 4000 to 7000 steps/day. The patient understood the rehabilitation and complied to participation in all sessions. During the duration of the program, there was no increase in Riluzole and NSAIDs that were prescribed in the past, and no additional medication was administered during hospitalization.

### 2.4. Follow-Up and Outcomes

One week after starting the SHAiR program, the patient experienced a change in his own gait posture and subsequently noticed an improvement in the symptoms. The patient’s symptoms and outcome change were reassessed after 2 weeks of treatment. The patient’s standing posture improved significantly after the two-week program (Figure 1c,d). In addition, it has become possible to walk with the head raised (Appendix A). The ALSFRS-R score changed from 38 to 37 points, and no definite functional decline was observed in limb muscle strength, speech, and swallowing. The muscular strength of the neck extension was improved to grade 3 in MMT. The results of the patient’s reported outcomes (PROs) are shown in Table 2.

Neck pain was gradually relieved during the first week and eventually improved from 9 to 6. Neck dysfunction indicated by the neck disability index (NDI) improved from 23 points to 10 points. The health-related quality of life (QOL) assessed using the 36-Item Short-Form Health Survey version 2 (SF-36) acute version were improved in all subscales except for bodily pain. No adverse events such as the appearance of myalgia and/or in-creased fasciculations were observed during the follow-up period. The patient’s treatment process is summarized in Figure 3.

## 3. Discussion

DHS is secondary to primary diseases such as PD, MG, and ALS, with the exception of idiopathic DHS [2]. According to previous studies, these primary diseases are categorized into neurological, neuromuscular, muscular, and other causes, and ALS belongs to the neurological type [6]. The proportion of ALS patients among DHS patients has been reported to be approximately 7% [2]. In addition, two relatively large cohort studies reported that the proportion of patients with DHS in ALS was 1.3% (9/683) and 2.9% (3/105) [3,4]. Therefore, ALS with DHS is a very rare condition. DH due to ALS can be caused by the loss of cervical motor neurons [7]. However, clinical symptoms and natural history differ among patients [3,4], and the detailed mechanism of developing DHS has been unclear.

Orthoses and rehabilitation are the first-line conservative therapies for DHS [8]. However, reports on ALS with DHS are limited to orthoses. Cervical orthosis keeps the cervical spine in the extended position and improves problems associated with neck discomfort, pain, eating, and social interaction [9,10,11]. However, its use has been reported to cause some problems such as discomfort and limitation of movement [12,13]. In addition to idiopathic DHS patients, rehabilitation treatment outcomes for PD, multiple system atrophy, cervical spondylosis, cervical spinal cord myositis, and post-radiotherapy patients have been reported [5,14,15,16,17,18,19,20,21,22,23]. There is no consensus on the effect of various treatment programs in rehabilitation that includes physical therapy, combined physical therapy and orthoses, hybrid assistive limb (HAL), chiropractic or athletic rehabilitation-based physical therapy, and the SHAiR program [5,14,15,16,17,18,19,20,21,22,23]. The SHAiR program was the only treatment in which the efficacy of treatment was evaluated by comparison, and good short-term results have also been reported [5]. In addition, it is highly useful since it consists of a simple and standardized training program that does not require expensive therapeutic equipment and can be easily reproduced by other therapists.

In this report, an ALS patient with DHS showed improvement in symptoms after undergoing the SHAiR program. The concept of the SHAiR program is to improve the function of the cervical extensor and flexor and global spinal alignment [5]. The patient showed improvement in cervical extension muscle strength. Muscle weakness due to ALS is affected by disease progression and disuse [24]. Kato et al. reported that strength training for ALS may improve disuse muscle weakness [25]. The SHAiR program includes cervical extensor muscle strengthening exercises, including cervical paraspinal muscles exercises. Therefore, the SHAiR program may have improved the disuse muscle weakness of the patient’s cervical extensor muscles. The patient also showed improved DH during standing and walking. The SHAiR program includes systemic exercise and walking training. Interestingly, Miura et al. reported a case that demonstrated improvement by walking training alone using HAL [20,21]. Although the detailed mechanism of these improvements is unknown, the whole-body approach that includes walking training may have affected the improvement of DH. In addition, the patient in this report showed improvement in most of the PROs, including the degree of cervical pain. The improvement of cervical pain by the SHAiR program has been substantiated by the effect of deep cervical flexor muscle exercise [5], and a similar mechanism may have improved the patient’s neck pain. The relevance between the SHAiR program implementation and changes in the SF-36 subcomponents should be carefully interpreted. This is because the factors that could explain the change in scores such as depression [26] and cognitive and behavioral symptoms [27] were ignored. In addition, the results were immediate and the patients’ living environments were different before and after the assessment. Therefore, to judge this association as beyond the realm of chance is difficult at this time. However, considering that (1) DH and related symptoms of this patient were improved at the timing of implementation of the SHAiR program instead of conventional rehabilitation, (2) the relevance of NDI and SF-36 has been described in patients with neck pain [28], and (3) the effect of exercise on SF-36 in general ALS patients is unclear [29], the change in SF-36 scores could also be interpreted as a response to improvement in DH and related symptoms and/or improvement in neck disability. For this discussion to move forward, the related factors for QOL in ALS patients with DHS need to be studied in detail, including immediate and long-term results.

Although this report presents a successful example of the SHAiR program, there are some notable points to consider regarding its implementation. The first is safety. Previous studies have reported that exercise for ALS does not accelerate the progression of symptoms [30]. Therefore, the SHAiR program for ALS probably pose a low risk of affecting the patient’s prognosis. However, Meng et al. reported the occurrence of adverse events such as myalgia, increase of fasciculations, nocturnal cramps, extreme fatigue, and restless legs as a result of exercise in patients with ALS [30]. Although the patient in this report did not show these symptoms during treatment, careful consideration should be given to the appropriate load. The second point of consideration is the indication for treatment. ALS with DHS is associated with motor neuron shedding and can be exacerbated in parallel with disease progression. Patients with ALS entering the terminal stage present with severe respiratory distress. In these cases, the exercise based SHAiR program may not be applicable. In addition, the approach to the whole-body that include walking training is considered to be one of the reasons why the SHAiR program was successful. Although most ALS patients with DHS maintain independent gait until the end of life, some patients are bedridden [3,4]. Therefore, the SHAiR program may not work sufficiently for patients with marked weakness of the lower limbs and patients who have difficulty walking independently. Further research is needed to determine what patients the SHAiR program is indicated for. The last point of consideration is effectiveness. Although this report presented good short-term results, there are limitations to the interpretation of efficacy, because the long-term results remain unknown. However, it is possible that the SHAiR program was effective in improving this patient’s symptoms, considering the lack of improvement that was observed with conventional rehabilitation. In the future, further research is needed using multiple samples with long-term results.

## 4. Conclusions

There is no established treatment for ALS with DHS due to the rarity of the disease. Our report suggests the potential efficacy of the SHAiR program for this patient. Further research on the efficacy, safety, and indications of the SHAiR program needs to be developed in the future.

## Figures and Tables

**Figure 1 medicina-58-00452-f001:**
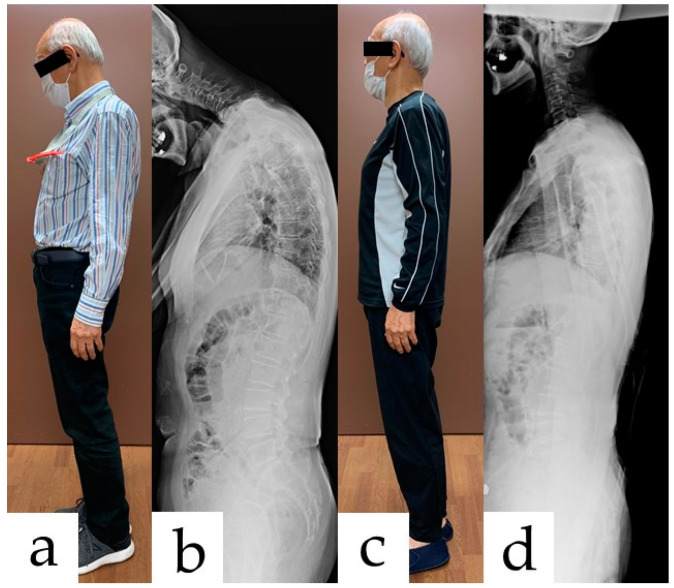
Pre and post-rehabilitation posture. Before rehabilitation: (**a**) lateral view of the photograph and (**b**) standing radiograph. Two weeks after rehabilitation: (**c**) lateral view of the photograph and (**d**) standing radiograph.

**Figure 2 medicina-58-00452-f002:**
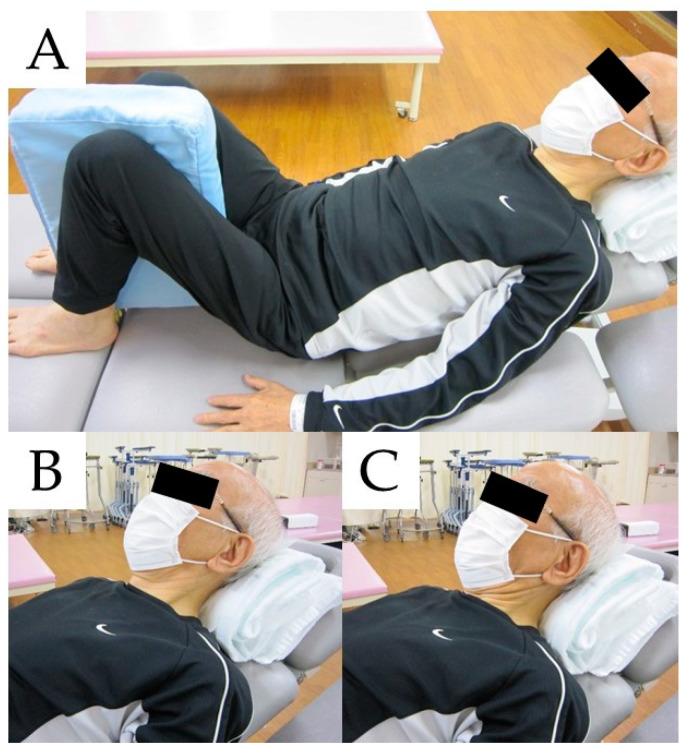
Posture during exercises. (**A**) Training posture, (**B**,**C**) deep cervical flexor muscle exercise.

**Figure 3 medicina-58-00452-f003:**
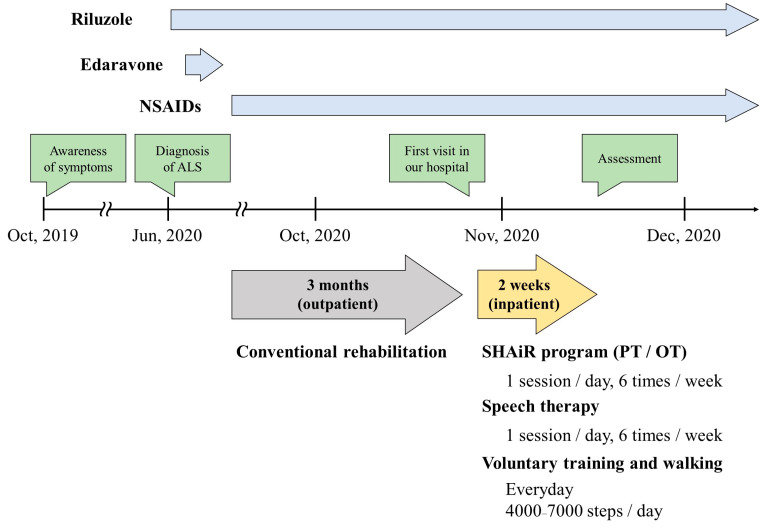
Timeline of patient’s treatment process. Abbreviation. NSAIDs, non-steroidal anti-inflammatory drugs; ALS, amyotrophic lateral sclerosis; SHAiR program, the short and intensive rehabilitation program; PT, physical therapist; OT, occupational therapist.

**Table 1 medicina-58-00452-t001:** Patient’s Clinical Findings.

Physical Examination
Upper motor neuron signs Increased jaw jerk reflex Increased deep tendon reflexes of the upper and lower limbs
Lower motor neuron signs Weakness and muscular atrophy of the upper limbs and trunk
** Electrophysiological Examination **
Electromyography: Neurogenic changes present
Repetitive nerve stimulation: No decremental response
Nerve conduction study: Normal
** Blood and Cerebrospinal Fluid Examination **
Anti–AchR antibodies and anti–Musk antibodies: Negative
Cerebrospinal fluid: Elevated neurofilament light chain

**Table 2 medicina-58-00452-t002:** Patient-Reported Outcomes.

	Before Rehabilitation	2-Weeks after Rehabilitation
NRS		
Neck pain	9	6
NDI (points)	23	10
SF-36		
Physical functioning	30.0	75.0
Role physical	6.3	12.5
Bodily pain	12.0	12.0
General health	35.0	57.0
Vitality	25.0	68.8
Social functioning	0.0	25.0
Role emotional	0.0	8.3
Mental health	25.0	75.0

Abbreviations: NRS, numerical rating scale; NDI, neck disability index; SF-36, 36-Item Short-Form Health Survey version 2 acute version.

## Data Availability

Not applicable.

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
