# Peer review of "The Short and Intensive Rehabilitation (SHAiR) Program Improves Dropped Head Syndrome Caused by Amyotrophic Lateral Sclerosis: A Case Report"

_medicina, 2022, doi:10.3390/medicina58030452_

Round 1

Reviewer 1 Report

The paper is the first case report describing the improvement of both dropped head and cervical pain after the short and intensive rehabilitation (SHAiR) program in an ALS patient with dropped head syndrome.

As the authors describe in the discussion, the appropriate load of the training needs to be carefully considered. For the readers, I would like to ask the authors to provide detailed information about the duration, the number of repetitions, and strength of each of the training menu of SHAiR program including the ones presented at line 117-120, throughout the 2 weeks of training period.

Reviewer 2 Report

Interesting and important clinical article due to its rarity and difficulty in adequate treatment.

However, the rehabilitation program should be more descriptive.

Reference 27 in line 223, should be confirmed.
